# Using the DREAM Methodology for Course Assessment in the Field of ICT-Enabled Education for Sustainability

Vassilios Makrakis [1,2]

1   School of Education and Social Sciences, Frederick University, 7, Y. Frederickou Str., Pallouriotissa Nicosia 1036, Cyprus; makrakis@edc.uoc.gr
2   Department of Primary Education, University of Crete, 74100 Rethymnon, Crete, Greece

**Abstract:** This study explores the application of the DREAM methodology for course assessment in three South East Asian universities aiming to embed sustainability and sustainable development goals (SDGs) in multiple academic disciplines enabled by information and communication technologies (ICTs). A mixing of content and thematic analysis was used, which aligns with the underpinning philosophy of the Diagnosing, Reviewing/Reflecting, Explaining, Assessing, Managing (DREAM) methodology. The DREAM methodology integrates five processes, starting from diagnosing, to reviewing/reflecting, explaining, assessing, and, finally, managing. Results show that merging semantic and latent themes has contributed to uncovering what messages students' narratives convey and provided a space for focusing both on the surface and explicit meanings of the data as well as on theory building and policy making. They also show the effectiveness of the DREAM methodology in constructing new knowledge and generating meaningful interpretations and suggestions to teacher educators and other academic teaching staff, as well as higher education institutions' policymakers and planners.

**Keywords:** DREAM methodology; course assessment; South East Asia; DeCoRe plus; ICT; Education for Sustainability; SDGs; curricula

## 1. Introduction

One of the most visible side effects of Asia's rapid growth has been environmental damage. Recent climate-related disasters in the region show that Asian policymakers must act now to protect their citizens and mitigate and reverse the negative impacts of climate change to secure sustainable growth for the future [1]. A way to respond to these challenges is through Education for Sustainability (EfS), enabled by information and communication technologies (ICTs). Addressing complex challenges and current and future uncertainty is at the heart of Agenda 2030 and the 17 Sustainable Development Goals (SDGs). The link between ICTs in education and EfS is being addressed by extensive debates and research, which recognize the challenge new technologies bring to the reorientation of education towards learning to live together sustainably [2,3]. ICTs can help learners explore concepts, engage in problem-based and authentic learning, enhance meta-cognitive skills, and present information using multiple media [4–6]. The shift from Web 1.0 to Web 3.0 technologies provided learners with opportunities for teaching, learning, and social interaction, which is highly fostered by the open education movement and the creation of Creative Commons Licenses authorizing free access to knowledge, digital tools, and learning objects. All these are closely related to the goals, themes, and learning objectives addressed by Education for Sustainability (EfS), that can be enabled through ICTs. More specifically:

- EfS themes such as gender equality, poverty, child labor, climate change, and energy waste integrated into the school curricula could provide a worthwhile context for developing ICT knowledge and skills.

- ICT tools such as concept mapping, audio–visuals, gamification, and digitization can provide meaningful and challenging contexts for addressing a wide range of EfS themes and sustainable development issues.
- SDGs provide a worthwhile and meaningful context for addressing ICTeEfS. For example, SDG12 points to the importance of using technological resources in a responsible manner. SDG9 includes, among its specific goals, the urgent need to use technological resources efficiently, promoting clean and environmentally sound technologies. SDG13, for its part, calls on education to raise awareness and empower society to prevent the aggravation of climate change.

Although new technologies can play a key role in EfS by providing teachers with new tools that can transform and enrich instructional roles, curricula, and practices, technology by itself is not likely to contribute unless it is based on sound pedagogical principles [7,8]. Added to that, the integration of ICTs does not merely mean the addition of ICT as a subject or tool. It implies changes in teaching and learning and requires comprehensive and integrative planning. ICTeEfS requires a transformative, action-oriented pedagogy that promotes autonomous learning, participation, collaboration, problem solving, interdisciplinarity, and transdisciplinarity, and linking formal and informal learning [9–11]. ICTs are an essential component of the implementation of EfS, both in terms of pedagogy rethinking and developing teachers' capacities to address SDGs.

An ICTeEfS capacity-building project was initiated by a Consortium of South East Asian and European Union universities funded by the European Commission for the period 2019–2023. The project was co-ordinated by Frederick University (Cyprus) with the participation of the University of Crete (Greece), the Regional Center of Expertise in Education for Sustainable Development (RCE) Crete (Greece), two universities in Indonesia (Indonesian University Education, and Gadjah Mada University), three universities in Malaysia (Open University Malaysia, University Science Malaysia, and University Technology Malaysia), and two Universities in Vietnam (International University and University of Social Sciences and Humanities).

In the case of the ICTeEfS project, a six-stage capacity-building model for training academic staff and teachers to embed sustainability in their courses and teaching was adopted, as depicted in Figure 1 [12].

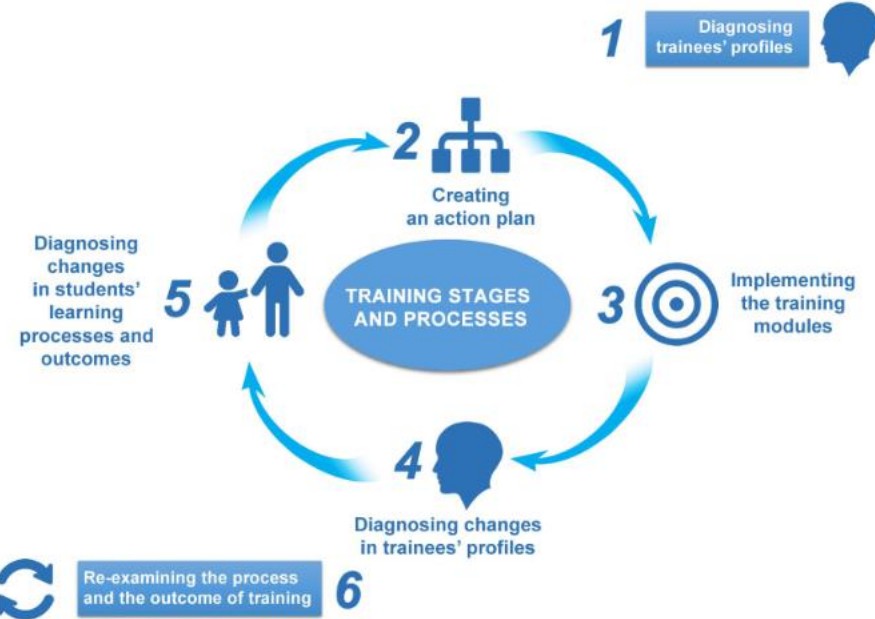

**Figure 1.** A capacity-building model [12].

The accelerated/cascading training model applied ended up with training 103 teacher educators and other academic teaching staff from the seven partner universities and

1451 in-service teachers who, in turn, provided in-house training that reached up to 910 trained teachers within a period of three months. The total amount of training was 727 h, and 722 digital lesson plans were developed as part of the training tasks addressing sustainability issues and SDGs. In the training workshops, particular emphasis was given to the participatory and negotiated curriculum development approaches, the six pillars of learning (learning to know, learning to be, learning to do, learning to live together sustainably, learning to give/share, and learning to transform oneself and society) [4,13] as well as the 10 Cs of transversal skills, including critical consciousness, critical reflection, connectivity, co-responsibility, cross-cultural understanding, and construction of knowledge [4].

For course revision to embed sustainability and, most specifically, the 17 Sustainable Development Goals (SDGs), the DeCoRe+ methodology, an abbreviation of Deconstruction–Construction–Reconstruction advanced by Makrakis [4], was applied. The Deconstruction–Construction–Reconstruction (DeCoRe+) is an approach that sees curricula as living texts by adopting a design process that focuses on transformative curriculum development. The six processes comprising DeCoRe+ include the following processes:

1. Diagnosing: Reflecting on: (a) who we are; (b) what we have (existing knowledge); (c) where we want to go; and (d) why we want to go there.
2. Deconstructing: Analyzing critically the functioning of personal perspectives and habits of the mind and chosen curriculum units/modules.
3. Constructing: Gathering resources, creating ideas, and constructing new meanings (perspectives).
4. Reconstructing: Integration of newly constructed knowledge in line with the reconstructed frame of reference.
5. Implementing: Carrying out the reconstructed curriculum unit/module supplemented by service learning.
6. Finalizing: Reflecting and evaluating on what has been learned and changed.

Each partner university revised 9–11 courses infused with sustainability issues largely drawn from the 17 SDGs and the local societies using the DeCoRe+ curriculum development methodology. The rate of revision was driven by internal institutional policies and reached up to 20% for a bit more than half of the total courses (56%). The sustainability themes infused in the revised course curricula represented a wide diversity of academic subjects and disciplines, as well as SDG themes such as climate change, gender, poverty, quality education, sustainable consumption, energy use, sustainable urbanization, water security, deforestation, sustainable agriculture, biodiversity, child labour, sustainable tourism, fair trade, and social justice. The revised courses were also enriched by digital content and ICT tools, especially in the case of student assignments. As pointed out earlier, ICTs were seen as learning technologies that could contribute both as course delivery tools and tools enabling the process of teaching, learning, and curriculum development.

The revised courses were attended by students within two academic years 2020–2022. In terms of the curricular area, most of the courses revised fall into the Education Sciences (31 courses), followed by Social Sciences and Humanities (17), Sciences and Engineering (12), as well as Environmental Sciences (5). There were also seven non-credited courses attended by all students regardless of academic discipline; 80% of the 90,897 students attended these courses. In this study, the focus is on a formative assessment that was carried out in the middle of the implementation process using the Diagnosing, Reviewing/Reflecting, Explaining, Assessing, Managing (DREAM) methodology.

The overriding aim of this article is to present the DREAM methodology and the way it has been used to embed sustainability and SDGs in course curricula. The specific objectives are: (1) present the stages of the DREAM methodology in line with its underpinning theory; (2) explore the implementation of the DREAM methodology in course curriculum revision by applying a combination of content analysis and thematic analysis; and (3) discuss the implication of the DREAM methodology as an innovative approach of transformative course assessment.

## 2. Methodology

### 2.1. The DREAM Data Analysis Design

In this study, content analysis and thematic analysis were used for data collection, analysis, and interpretation of the results. These two methods are used to uncover themes in textual data, while content analysis can be either a quantitative and/or a qualitative approach that also involves thematic analysis [14–16]. The methodological framework advanced in this study (Table 1) attempted the mixing of semantic and latent levels of themes where the first level focuses on the surface but explicit meanings of the data and the second is looking for anything beyond what a participant has said or what has been written [15]. Thus, the analysis applied in this study moves beyond describing what is said and focuses on interpreting and explaining data, along with identifying or examining the underlying ideas, assumptions, ideologies, epistemologies, and perspectives that are shaping or informing the semantic content of the data. This type of analysis is in line with the transformative assessment [17] and the Diagnosing, Reviewing/Reflecting, Explaining, Assessing, Managing (DREAM) methodology for course curriculum assessment.

**Table 1.** The DREAM data analysis design.

| Processes | Description |
| --- | --- |
| Familiarization | Reading and re-reading the data transcripts to familiarize myself with the entire body of data collected, generating first codes, and eliciting impressions. |
| Initialization | Organizing the data in a more meaningful and systematic way for generating more refined codes that help to reduce the huge amount of data into small chunks of meaning and elaborate further codes and meaningful outcomes (initial themes). It also refers to the process of generating ideas and making sense of data, depending on researchers' familiarization with data through immersion. Immersion is achieved through careful reading of transcripts, and listing meaningful, recurrent ideas and key issues in data. |
| Determination | Determining coding categories and searching for meaningful themes consisting of codes fitting into a theme shifts the level of analysis to capturing patterns that reflect significant and interesting incidents. Classifying, comparing, and labeling are processes used to identify meaningful patterns in the data. |
| Verification | Avoiding bias and increasing the trustworthiness of the interpretations can be ensured through mechanisms such as reflective feedback (going back to the sources of data) and postponement of an in-depth literature review after the determination process. |
| Theorization | Theorizing refers to a level of abstraction through making convincing inferences to explanations grounded in data. Checking the applicability of the findings to other contexts, and generalizability, the transferability of the findings to other settings. |
| Presentation | Communicating the processes and practices in carrying out the mixing of content analysis and thematic analysis should be a straightforward storyline, avoiding adding too much information that can hide the key and meaningful findings. |

The soundness or the overall quality of this research is judged by its trustworthiness which is a term that is based on four criteria: credibility, transferability, dependability, and confirmability of research [18,19]. First, credibility was ensured by: (1) asking the opinions of key experts after developing each instrument; collecting in-depth data both through the DREAM instrument and interviews; prolonged engagement with data collection and analysis [20,21]; and, lastly, reflective feedback [22,23], that is, going back to the key data teaching staff who have implemented the revised courses and collected the data to validate and verify the results, inferences, and interpretations. Besides using interviews to generate a deeper understanding of respondents' experiences and beliefs towards the courses, they were also used for verification purposes. Second, to assure transferability, the 343 participants were selected through purposive sampling [24,25]. This is due to the fact that the target students were those who attended a number of the courses which have been revised to embed sustainability issues elicited from the 17 SDGs.

Most of the students were from the first and third year of their study programme, representing multiple academic disciplines. An open-ended Google form of the DREAM methodology instrument was distributed to the students. Once the data collected have been transcribed, search for relevant codes. The coding process involved identifications of issues,

similarities, and differences that were revealed through the data collected and students' feedback during the focus group interviews. Third, dependability and confirmability were addressed by an audit trail through which three experts responsible for the implementation of the revised courses monitored all stages of the study to establish that the study revealed objective results [25,26].

### 2.2. The DREAM Methodology Stages

The DREAM approach adopted focuses on assessment for learning that involves teachers using evidence about students' knowledge, understanding, and skills to inform not only their teaching but also the content of learning. Sometimes this is referred to as 'formative assessment' since it usually occurs throughout the teaching and learning process. This contrasts with the assessment of learning, usually referred to as a "summative" assessment, taking place at the end of the course or unit teaching. Figure 2 summarizes the stages that constitute the DREAM course assessment methodology.

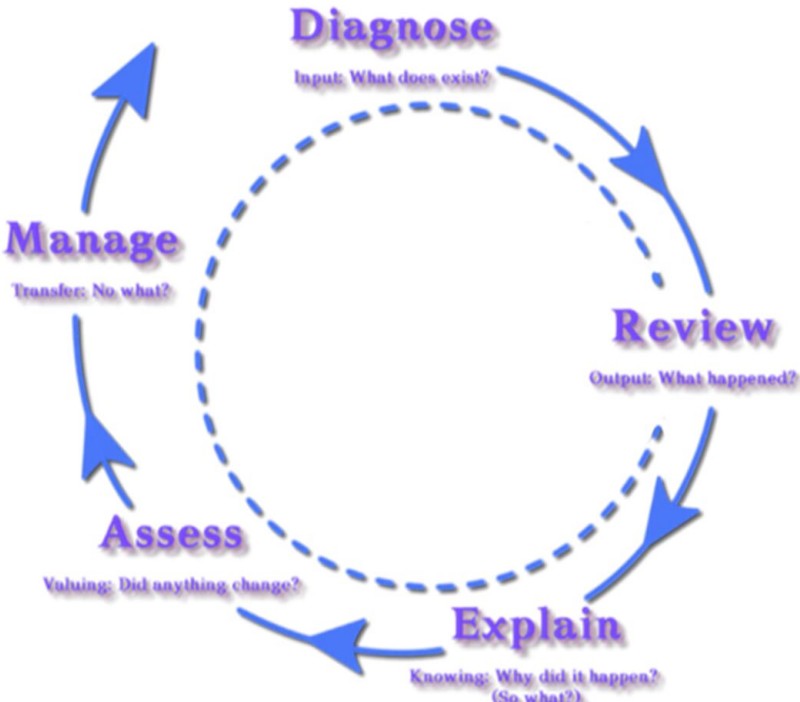

**Figure 2.** The DREAM methodology stages.

Diagnosing refers to the process used for identifying students' level, background, and pre-existing knowledge of the subject matter. It is a form of pre-assessment that provides opportunities to reflect on students' thinking, strengths, weaknesses, expectations, and perspectives. Reviewing/reflecting refers to the process of uncovering positive and negative incidents during the course implementation, what students expected and what was different from their expectations, unexpected things taking place, students' reactions due to differences in their expectations, and suggestions that could help to improve the course. This process asks the question: What happened? What did you do? What did you expect? What was different? How did you react? What did you learn? In the process of explaining, the emphasis is placed on interpreting the findings revealed from the questions asked in the previous process. For example, in the case where some of the expectations for the course have not been met, there is a need to elaborate on why this is so and the reasons behind that. Assessing focuses on what students have learned during the course implementation that may contribute to their: (1) knowledge construction which, in turn, will possibly raise employability opportunities; (2) civic engagement, turning them into agents of change; (3) living together sustainably; and (4) personal development. This

process asks the following questions: Why does it matter? How do your experiences relate to what you learned in this course? How do your experiences link to your academic, professional, personal, and/or civic engagement and development? Finally, the process of managing refers to personal and institutional perspectives. In particular, the emphasis here is placed on eliciting data on what can be changed and/or how the learning gains can be applied in other contexts. Similarly, it concerns the changes needed as a reflection of the experiences gained, not only in terms of those involved in the process but also in terms of course curriculum development and institutional policy. The questions asked here are: What would you keep, change, or do differently next time? How will you apply what you have learned to your future career/life? What does it mean to you and/or what suggestions do you have about how course content and methods might be improved? How can you accomplish your personal or collective responsibilities within the system or challenge the system? What can you change or how can you apply the new learning of the course in the future?

## 3. Results

### 3.1. Diagnosing

Table 2 shows that 81 students out of 343 (23.9%) declared that they had attended a similar course, and an equal percentage for those who had created a similar project to the courses attended. However, the percentage of students who ever participated in a work dealing with issues similar to the topic of this course compared with the ones who have never performed similar work dropped to 17%. Concept mapping was used to identify students' previous knowledge of the content of the courses attended and the results showed that the great majority of the students exhibited very limited knowledge and, in some cases, that knowledge lacked meaningful relationships.

**Table 2.** Students' responses.

| Answers | Attended Similar Course | | Created a Project Similar to the Course | | Worked Dealing with Issues Similar to the Course | |
|---|---|---|---|---|---|---|
| | Students | % | Students | % | Students | % |
| YES | 81 | 23.5 | 72 | 23.5 | 58 | 17.2 |
| NO | 262 | 76.5 | 233 | 76.5 | 279 | 82.8 |
| Total | 343 | 100.00 | 305 | 100.00 | 337 | 100.00 |

Results on student's expectations from the courses implemented can be categorized into five main themes, namely: (1) familiarization with the course and the context, (2) personal improvement, (3) professional development, (4) civic engagement, and (5) expanded networking or socializing with peers as the following narratives show.

*". . . can learn how to socialize with the community. . ."*
*". . . can learn how to make good programs for the community. . ."*
*". . . improve skills on communication, teamwork, and problem-solving. . ."*
*". . . can know more about the real problems in the community. . ."*
*". . . can give a better understanding of what the community actually needs. . ."*

As for what they could give to the community, findings revealed that they expected to find themselves beneficial to the community.

*". . . be able to implement the knowledge from university into the community. . ."*
*". . . be able to empower the community. . ."*
*". . . be able to apply relevant skills. . ."*
*". . . can give something meaningful to the community. . ."*
*". . . provide a social service to the communities who need it the most. . ."*
*". . . can give ideas and positive impacts to the community. . . ."*

A considerable number of students' responses showed that they expected to achieve an enhancement in their skills and be responsible people in their daily life and career, as

well as be prepared for the future. In the case of professional development, there were two emerging themes: general and specific topics and knowledge on the course, and career prospects or employability. *"...I hope that this topic could be applied to my future career ..."; "I expect I will understand some important and controversial issues (climate change policy, nuclear power, nuclear power, recycling policy, and traffic congestion charging)".* A considerable number of students declared expectations concerning policies and practices, as in the following, *"... the impact of government policies on the environment and economics. How the government can balance and develop sustainably both economics and environment". "... policies that could impact the way people invest in the environment and more concepts related to Environmental economics." "I look forward to policies that successful entrepreneurs need to follow or ensure to run a business. "I think it would be great to learn more about the connection among practices in environmental science, economics, and politics field."* Some students expected the teachers can make the topic easy to understand and easier to remember, so they can complete the courses successfully. Another expectation from students for the courses was the opportunity to get to know other students from different faculties. Thus, they can expand their networking for professional development, career, and personal development.

### 3.2. Reviewing/Reflecting

Table 3 describes the key themes revealed that show positive and negative aspects. There are a number of similarities among the students in the three countries such as peer-to-peer collaboration, soft skills development, and new and deepened knowledge construction. Obtaining soft skills was a recurring statement by students, such as: *"collaboration, managing time, public speaking, knowing self, punctuality, reflecting, sharing, giving, and connecting". "... developed my public speaking skills in front of people despite it being only an online course"; "I became more adventurous in expressing opinions and applying critical thinking to analyzing a case I learned about how to manage my time". "I can appreciate time more, manage time properly, and do everything systematically and thoughtfully". "I became aware of myself and what things I need to do in the future".*

**Table 3.** Positive and negative aspects.

| Country | Positive | Negative |
|---|---|---|
| Indonesia | Fun and enjoyable course<br>Peer-to-peer collaboration<br>New knowledge acquired<br>Focus on real-life issues<br>Socialization among peers<br>Well-designed sessions<br>Soft skills developed | Low Internet connection<br>Low level of social interaction<br>Cognitive overloading |
| Malaysia | New knowledge acquired<br>Focus on real-life issues<br>Socialization among peers<br>Soft skills developed<br>Better and deeper knowledge | Management problems<br>Low level of social interaction |
| Vietnam | Easy to understand, interesting presentation, have some practice and exercises | Low Internet connection<br>Cognitive overloading |

Despite courses being delivered online, students had the opportunity to make friends and work together. "I've been able to communicate and make new friends, widening and broadening my connections with people during this pandemic". "I met friends from various faculties and regions accompanied by a co-facilitator to hold a class that was fun and enjoyable in understanding the material given". It thus seems that there is a close correspondence between what they expected and what they experienced during the implementation of the revised courses addressing SDGs. Another theme revealed was related to the multi-disciplinary approaches applied in some courses, especially the ones that

attracted students from different disciplines. The function of co-facilitators in the case of the Indonesian university seemed to enhance further students' positive remarks. As the following narrations show, "My senior as a co-facilitator also was so amazing and she is so friendly and funny. It was like I had my first family from this course". "During the half period of this course, I got so many experiences with my new friends . . . and I feel so happy, although it was just via online it was so enjoyed".

The results also show that the courses implemented were driven by an appropriate e-learning strategy with well-supported materials, although most of the students were not in favor of online learning. *"The teaching method and the website provided was helpful to help me understand the topic better". "The material coverage was complete and they discussed various topics that I find are important for the students to understand". ". . . the course is very well-prepared and . . .fascinated that the course was well planned and the content was so interesting".* Another positive impact usually mentioned by students was that of opening students' horizons by providing opportunities for obtaining new insights, as the following answer indicates: *"I'm very excited to follow this course. I am grateful for it. This course also gave me a lot of experience and insight".*

The factors that negatively affected students' teaching and learning processes during the implementation of the revised courses can be classified according to their importance, as follows: (1) technical; (2) teaching delivery mode; (3) content overload; and (4) interaction. From the technical part, the most frequently cited problem was the disruption of the online classes due to low Internet connectivity and quota. The online delivery mode of instruction was received as a necessity due to the COVID-19 pandemic, but not as a preferable solution. The most frequent reason was the lack of socializing directly with people. Statements such as *"I feel sad because I can't feel the euphoria and can't meet people directly"; "Eye fatigue and headache due to long screen time attending the course and doing the assignment".* It seems that this problem was related to the volume of the students' tasks and assignments which created anxiety and stress due to the cognitive overload. All these problems seemed to limit interaction and socializing, which are highly valued in South East Asian societies.

In general, students did not declare any differences from what they expected at the diagnosing stage, and the differences mentioned were mostly related to the expected complexity of the revised courses addressing SDGs, but most declared that the course could convey the knowledge in understandable ways. A number of students who expected that the course would be mainly based on theory, and not the practical aspect, experienced that the courses were practice-oriented.

Suggestions included: more examples, quizzes, showing some practical videos or case studies which are familiar to them, uploading the assessment results to online systems so they know where they were wrong, having more group discussions, and higher interaction between the lecturer and the students. They also suggested that the course notes should be uploaded before the class begin. Some thought that they have to prepare the lesson carefully, read and understand clearly the key readings, then they need to review the lesson through the content of the slides, concept maps, and flashcards on the Internet many times after the lectures.

### 3.3. Explaining

In the process of explaining, the emphasis was placed on justifying the findings revealed from the questions asked in the previous stages. From the analysis, it has been revealed that most of the students connected the negative issues with the online mode of teaching due to the COVID-19 pandemic. Despite the popularity of online learning during the pandemic period, most of the students' critical incidents are related to this mode of instruction. Southeast Asian students are highly social people and social interaction was an issue during the online classes.

In general, social interaction, whether it is taking place in face-to-face classes or online classes, can lead to the students' satisfaction, increased engagement, and motivation. The problem with students' negative perceptions is largely attributed to problems with Internet

connectivity that caused a distraction, and the implementation of the mode of instruction found both students and academic teaching staff unprepared.

There is no evidence from the students' perspectives expressed in their narratives that online teaching and learning do not offer opportunities for meaningful and sustained social interactions. Looking specifically at how online courses are delivered, there is evidence to assume that even the course content was not appropriate to the remote delivery of instruction and the online learning environment. Perhaps the demand of students for more flexible learning materials such as digital tools, case studies, and real-life applications partly explains their concerns.

### 3.4. Assessing

The process of assessing focused on what students have learned during the course implementation that may contribute to their: (1) knowledge construction which, in turn, will possibly raise employability opportunities; (2) civic engagement, turning them into agents of change; (3) living together sustainably; and (4) personal development. The assessment was categorized into four categories: (1) impact on students' academic development; (2) impact on students' professional development; (3) impact on society; and (4) impact on students' personality. The concept maps created at the end of the courses were significantly improved compared with the concept maps created at the diagnosing stage. The aggregated assessment results are presented in Table 4.

**Table 4.** The key finding in terms of impact on students.

| Impact on Students' | Key Findings |
|---|---|
| Academic development | <ul><li>Increased knowledge of social and environmental issues.</li><li>Ability in merging knowledge and practice focusing on SDGs.</li><li>Skills in using analytical tools to tackle sustainability issues.</li><li>Discuss positive and negative externalities, and multiple perspectives/worldviews.</li><li>Improved knowledge in interpreting and analyzing data.</li><li>Filtering information from multiple sources.</li><li>Increased knowledge of teaching methodology and curriculum design to address SDGs.</li><li>Increased knowledge to contextualize ICTs with SDG themes and SDG themes with ICTs.</li></ul> |
| Professional development | <ul><li>Knowledge and skills acquired raised students' expectations for better career development.</li><li>Competencies to work in teams, time management, communication, reasoning, creativity and innovation, self-confidence, multitasking, and empathy.</li></ul> |
| Personal development | <ul><li>There was an overlapping of statements related to professional and personal development. Some distinct ones were the following:</li><li>Attitudes cultivated included: cheerfulness, affection, open-mindedness, confidence, responsibility, ethics, self-esteem, and loyalty.</li></ul> |
| Society | <ul><li>Civic/political engagement, efficacy, and advocacy related to SDGs through practicum driven by service-learning or community-based learning.</li><li>Applying theory, reflection, and action toward building a more sustainable and just society.</li><li>Appreciating the importance of ethics in dealing with sustainability issues.</li><li>Ethical entrepreneurship and active citizenship.</li></ul> |

### 3.5. Managing

The process of managing refers to personal and institutional perspectives. In particular, the emphasis here is placed on eliciting data on what can be changed and/or how the learning gains can be applied in other contexts. Similarly, it concerns the changes needed as a reflection of the experiences gained, not only in terms of those involved in the process but also in terms of course curriculum development and institutional policy. The analysis of the data reveals that they are highly practical-oriented and, thus, they mostly favor courses that address real-life issues that reflect their own experiences and social contexts. Looking at expanding their opportunities for future careers/life, and accomplishing their personal

targets, students provided a series of suggestions that could help improve not only the revised courses but also the quality of education addressed by SDG4. Thus, in answering the question of what they wanted to change, or how they best apply the new knowledge, skills, attitudes, values, and action competencies learned from the revised courses they attended, the following are included:

- Be more careful about environmental degradation and human-nature's impact.
- Apply knowledge to the real world, change points of view, and have a better look at life.
- Work and function in a more sustainable way.
- Merge knowledge, reflection, and action.
- Do business through the ethical lens.
- Analyze the interdependencies of the four (environment, society, economy and culture) sustainability pillars.
- Integrate the policy perspective in sustainability pillars.
- Evaluate alternative ways of doing things in dealing with sustainability issues.
- Raise awareness of protecting the environment and promoting people's well-being.
- Be driven more by sustainability ethics than economic profit.

If they would take the same course again, they would like the following changes:

- Increased student–teacher and student–student interaction.
- Integrate community-based learning and practicum placements.
- Combine the system of activities online and offline.
- Integrate more digital learning materials and tools as well as social media in teaching, learning, and curriculum.
- Integrate more hands-on activities, especially field visits.
- Apply problem-based and project-based activities addressing SDGs.

## 4. Discussion

This study explores the assessment of six courses included in this study that have been revised to embed SDGs enabled by ICTs with the participation of 343 students mainly in the first and third year of their undergraduate studies in three South East Asian universities (Indonesia, Malaysia, and Vietnam) in the period 2020–2022, applying the DREAM methodology. During the entire process, the academic instructors were challenged to achieve the rigor and credibility that make the assessment results be as trustworthy as possible. To validate the outcomes and to strengthen the validity of the data collected and the interpretations derived, a reflective assessment was performed, that is, going back and forth to the respondents and key investigators, not only as sources of data but also as validators of the interpretations. Although this approach does not support the replication of the outcomes, because the data arise from a specific context, the interpretations derived from the data can be transferred to other contexts and inform decisions about possible changes both for the teaching and learning process as well as the revised course content.

Results on students' expectations from the courses implemented were categorized into five main themes, namely: (1) familiarization with the course and the context, (2) personal improvement, (3) professional development, (4) civic engagement, and (5) expanded networking or socializing with peers. It has been also revealed that a considerable number of students' responses are they expected to achieve an enhancement largely in soft skills, and they are empowered to act as change agents through developing civic engagement enabled by service learning focusing on SDGs.

Expectations varied according to the content of the course. However, the general trend shows that students expected to acquire new and deep knowledge in the course subject area and how the new knowledge will be merged with practice and the real world. Another issue in terms of expectations was that an emphasis should be placed on knowledge related to processes rather than outcomes. In this context, there was a recurring finding of connecting the subject matter with social, environmental, and cultural factors. For example,

they expected to have a deeper understanding of how the natural environment changes when being manipulated by humans and mentioned that the content should focus on providing solutions to relevant questions. Other expectations were related to the content of the courses that should be enriched with digital applications and materials.

Communication and peer-to-peer collaboration were highly pointed out by a great majority of students during the course implementation. Students seemed to enjoy working with and supporting their peers during learning tasks in the courses attended. Peer-to-peer collaboration is an intended and planned teaching methodology and a formative assessment strategy, that can help to activate students as learning resources for one another [23,27]. This strategy is of critical importance for the work environment and, since the great majority of students connected their learning gains with employability, it makes it even more important. Students' narratives convey messages that support collaboration and collaborative learning in a wide area of settings, especially in merging the 3 Hs, that is the Head (cognition), the Heart (sentiments, feelings, and values), and the Hand (action and reflection), that lead to praxis [28].

Other themes that emerged were related to the multi-disciplinary approaches applied in the revised courses addressing SDGs, especially the ones that attracted students from different disciplines and the prioritization of soft/transversal skills development. Transversal skills and soft skills are usually considered interchangeably and can be transferred from one context (school) to another context (workplace) [29–32], and foster sustainable development [33]. The revised courses implemented integrated the 10 Cs—a transversal skills framework advanced by Makrakis [3]. Examples of soft or transversal skills prioritized by students included largely interpersonal skills such as time management, organizational skills, teamwork, self-discipline, enthusiasm, empathy, perseverance, and self-motivation. From the 10 Cs framework of transversal skills, critical thinking, collective responsibility, cross-cultural understanding, and constructing knowledge were highly mentioned by students. Cultivating students' soft or transversal skills from an experiential and transformative learning perspective is of paramount importance not solely for raising students' employability opportunities, but also for civic engagement and active citizenship. Connecting transversal skills with real-life issues elicited from the 17 SDGs was also highly cited by students across all subject and discipline areas in the three countries. Working with real-life issues, especially in times of community practicum placements or internships can make SDGs and sustainability issues more tangible and meaningful to students.

Students' suggestions were geared towards the need to enrich the teaching, learning, and curriculum of the revised courses with more real-life examples and case studies that merge theory, reflection, and action, as well as the local and global societal context. Enriching the revised course curricula with more real-life examples can stimulate the 10 Cs (transversal skills) and the need for an inter- and multi-disciplinary approach to problem solving. Prioritizing solutions to environmental and social problems was highly appreciated by the participating students in the study and integrating real-life examples in the revised courses seems to be a good strategy for enabling solutions to real-life problems. Such a strategy may also promote civic engagement since real-life issues elicited from SDGs tend to be meaningful and applicable to students' lives, either directly or indirectly.

Another issue raised was that of cognitive overload, a situation where students experience too many tasks at the same time in a course. Managing cognitive overload is crucial in the teaching and learning process and it has to also be tackled in the process of curriculum revision and development. The cognitive load theory that emerged from the work of Sweller [34] in the 1980s argues that it is of critical importance for the teaching practice to optimize students' cognitive load, by striking the right balance between too much and too little load [35]. Students demand more real-life examples and case studies as well as digital tools and applications, and expect to provide more room in the content of the courses. Thus, such resources could help to reduce abstract theoretical knowledge, which, in turn, could possibly reduce students' cognitive overload. Reducing cognitive overload facilitates students' working memories to process information, construct knowledge, and

be more creative [36]. Drawing also on students' pre-existing knowledge can help reduce cognitive load, to the extent that any knowledge distortions on the subject are corrected.

This study has revealed that online teaching during the COVID-19 pandemic was received as a necessity and not as a possible alternative to the teaching and learning processes and practices, despite the fact that a reasonable number of students declared that such a delivery mode was effective. However, their assumption was not clear on whether it is connected with their learning outcomes or if it is just an impression that facilitated their learning during a period of education in emergencies due to the COVID-19 pandemic [37]. On the other hand, students perceived the online delivery of the instruction as a barrier to socializing with other peers, but no evidence in relation to its learning effectiveness was detected. Through interviews, however, there was evidence to assume that a combination of face-to-face and online teaching and learning, what can be termed as blended learning, could be beneficial in today's modern classroom. This can be justified by the fact that blended learning allows learning to be more personalized and more easily accessed [38]. Blended learning can also respond to the student's demands for enhancing teacher–student and student–student interactions and collaborative learning or group work activities which fell behind in online classes. Previous research puts a strong emphasis on such interactions that drive students' learning outcomes [39]. Interpreting students' responses, it may be assumed that the online class interaction was not the one expected and the instruction was largely teacher-centered. Another dimension of interaction raised was connected to socializing with peers. It was clear that students demanded more social interaction that could possibly lead to increased collaborative learning. Sustaining teaching and learning interactions along with social interactions are crucial in online learning.

## 5. Conclusions

The ICTeEfS project's primary focus was on curriculum development through reconstruction to address Education for Sustainability enabled by learning technologies. To this end, developing participatory curriculum development to address sustainable development issues and SDGs was perceived as a process, context, and praxis, rather than a product as it is usually conceived and practiced. In this study, content analysis and thematic analysis were used for the data collection, analysis, and interpretation of the results with a mixing of semantic and latent levels of themes. Merging semantic and latent themes has contributed to uncovering what messages students' narratives convey and provides space for focusing both on the surface and explicit meanings of the data as well as theory building and policy making. This type of analysis is in line with the transformative assessment [17] and the theoretical foundations of the DREAM methodology for course curriculum assessment. The results of this study, besides exploring the application of the DREAM methodology for course assessment, can provide policymakers, educational planners, teacher educators, and other teaching staff with useful inputs that can inform the teaching and learning processes, and practices, as well as course curriculum development to address SDGs enabled by ICTs.

It is worth pointing out that there was an expectation to have more interaction with the students during the course assessment and especially during the verification process, but this was hindered by the restrictions posed due to the COVID-19 pandemic. It is, however, planned to transform the DREAM methodology into an online digital tool. Such a transformation will help to make this process more interactive, save time, and support obtaining meaningful feedback during the implementation process. Feedback from the learner and to the learner is a crucial element of the DREAM course assessment methodology, which can be better tackled through its digitalization.

**Funding:** This research was funded by the Erasmus+ Capacity Building in the Field of Higher Education European Commission, Project No. 598623-EPP-1-2018-1-CY-EPPKA2-CBHE-JP (2018-3774).

**Institutional Review Board Statement:** The institutional review board of the author's university of affiliation exempted the present secondary research of public-use data from review.

**Informed Consent Statement:** This research employed a public-use data set that does not contain identifiable information of participants. The original researchers that produced the data set obtained informed consent from the participants.

**Data Availability Statement:** The data is not publicly available due to privacy and ethical reasons. However, project information and relevant open accessible information is available at https://icteefs.frederick.ac.cy/.

**Acknowledgments:** This article is part of the dissemination of the ICTeEfS (ICT-enabled In-service Training of Teachers to Address Education for Sustainability). The content of this paper expresses the views of the author and the ICTeEfS Consortium and does not necessarily reflect the views of the European Commission. The European Commission is not liable for any use that may be made of the information contained herein. Thanks also to the respondents of this study, the academic staff, and Nanung Agus Fitriyanto at the University Gadjah Mada, Indonesia; Munirah Ghazali at the University Science Malaysia; and Pham Hoa at the Ho Chi Minh International University, Vietnam for co-ordinating data collection.

**Conflicts of Interest:** The author declares no conflict of interest.

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
