# Peer review of "Using the DREAM Methodology for Course Assessment in the Field of ICT-Enabled Education for Sustainability"

_ejihpe, doi:10.3390/ejihpe13070100_

Round 1

Reviewer 1 Report

The author should write the meaning of DREAM acronym when it is first mentioned in the abstract.

This study has revealed that online teaching during the Covid-19 pandemic was received as a necessity and not as a possible alternative to the teaching and learning processes and practices, despite the fact that a reasonable number of students declared that such a delivery mode was effective. H

The results of the study show that DREAM should serve as an important model for an on-line course. The manuscript is written in a clear and good way. It’s methodology is adequate, and the conclusions are in alignment with the findings. The focus of the course assessment is very important in developing literate citizens, e.g., civic engagement, or impact on society.

I agree with the author/s, that “The results of this study besides exploring the application of the DREAM methodology for course assessment, can provide policymakers, educational planners, teacher educators, and other teaching staff useful inputs that can inform the teaching, learning processes, and practices as well as course curriculum development to address SDGs.”

However, I have one minor comment:

The author should write the meaning of DREAM acronym when it is first mentioned in the abstract.

Author Response

First of all, I thank very much all the reviewers for the fruitful comments that have added to the quality of the article. The changes made are marked in red in the text. More specifically:

1ST REVIEWER

The author should write the meaning of DREAM acronym when it is first mentioned in the abstract. This has been done! Line 11

Reviewer 2 Report

Dear Author,

Thanks for your MS and the value added it brings us.

Nevertheless, there are some key issues that might be improved:

A. Title: I would recommend a slight adjustment in the MS´ title as follows, “Using DREAM methodology for Course Assessment Applied in ICT-enabled Education for Sustainability: A Southeast Asia experience”

1. In introduction:

In this area, I would advise defining and elaborating on the overall goal and the specific objectives.

There is, in my opinion, already a methodological introduction starting at line 63. Therefore, I'd like to offer a brief summary of what will be discussed next (2.1).

2. Material and methods:

I would recommend changing the name of this part to "methodology" and adding a subtitle, such as data collection and analysis (2.2) - line 123.

Please explain how you decided on a) purposive sampling and b) the method to employ for content analysis. in the two academic years 2020–2022, resolve the issue of choosing the first and third years.

4. Discussion

Given that this manuscript's events took place during the COVID 19 pandemic, I would advise including some "Limitations" directly.

5. Conclusions

I was hoping to read something regarding the following or future actions.

Best regards,

Reviewer

Author Response

First of all, I thank very much for the fruitful comments that have added to the quality of the article. The changes made are marked in red in the text. More specifically:

2ND REVIEWER

  1. Title: I would recommend a slight adjustment in the MS´ title as follows, “Using DREAM methodology for Course Assessment Applied in ICT-enabled Education for Sustainability: A Southeast Asia experience” The title has been changed according to the suggestion!

In this area, I would advise defining and elaborating on the overall goal and the specific objectives. This has been done! Lines 136-142

I would recommend changing the name of this part to "methodology" and adding a subtitle, such as data collection and analysis (2.2) - line 123. This has been done!

Please explain how you decided on a) purposive sampling and b) the method to employ for content analysis. in the two academic years 2020–2022, resolve the issue of choosing the first and third years. This has been done! Lines 170-171.

Given that this manuscript's events took place during the COVID 19 pandemic, I would advise including some "Limitations" directly. This has been done!  I was hoping to read something regarding the following or future actions. This has been done! Lines 525-532

Reviewer 3 Report

To the author

The article is well written and innovative giving a step forward in the Education area.

Small changes should be done to clarify some statements:

1) a clear description of research methodology, techniques, instruments and respondents. I suggest a table in the METHODS section. For example, mixed-method research supported in inquiry, field notes, artefacts...applied to teachers in training....

2) Sometimes participants and named students and other times teachers in training/ in-service. Although I understand the reason it may seem strange to readers not so familiar with teachers in training. Please explain it better along the text.

3) Concept maps are not mind maps. Please use the same words if you are using them as pre and post-instruments to compare the effect of the intervention. Another option is to clarify the difference between them and explain the reason why the diagnosis phase was evaluated with mind maps and the end phase with concept maps. ( I think the concept of the mind map is unappropriated use) 

4) Please better explain why ICTs are so referred, because I only read about online classes and that students/teachers in-service did not really like it. Clarify.

Good work and congratulation on an innovative approach.

Author Response

First of all, I thank you very much for the fruitful comments that have added to the quality of the article. The changes made are marked in red in the text. More specifically:

I suggest a table in the METHODS section. Changed line 143.

2) Sometimes participants and named students and other times teachers in training/ in-service. It has been made clear that the students are the focus of the paper while teachers in training/in-service refer to those who participated in the whole project.

3) Concept maps are not mind maps. The mind map is replaced to avoid confusion!

4) Please better explain why ICTs are so referred, because I only read about online classes and that students/teachers in-service did not really like it. Clarify. Reference to ICT relates to the content of the courses and the whole project. A notice has been made in lines 123-127.